# KEA Explain: Explanations of Hallucinations using Graph Kernel Analysis

**Reilly Haskins**                                    REILLY.HASKINS@PG.CANTERBURY.AC.NZ
**Benjamin Adams**                                 BENJAMIN.ADAMS@CANTERBURY.AC.NZ
*Department of Computer Science and Software Engineering, University of Canterbury, New Zealand*

**Editors:** Leilani H. Gilpin, Eleonora Giunchiglia, Pascal Hitzler, and Emile van Krieken

## Abstract

Large Language Models (LLMs) frequently generate hallucinations: statements that are syntactically plausible but lack factual grounding. This research presents KEA (Kernel-Enriched AI) Explain: a neurosymbolic framework that detects and explains such hallucinations by comparing knowledge graphs constructed from LLM outputs with ground truth data from Wikidata or contextual documents. Using graph kernels and semantic clustering, the method provides explanations for detected hallucinations, ensuring both robustness and interpretability. Our framework achieves competitive accuracy in detecting hallucinations across both open- and closed-domain tasks, and is able to generate contrastive explanations, enhancing transparency. This research advances the reliability of LLMs in high-stakes domains and provides a foundation for future work on precision improvements and multi-source knowledge integration.

## 1. Introduction

Despite the impressive capabilities of Large Language Models (LLMs), they frequently generate outputs that are grammatically correct but semantically incorrect—referred to as hallucinations (Huang et al., 2025). These hallucinations are a significant challenge, particularly for high-stakes domains such as healthcare, legal advice, and education, where misleading information can seriously degrade user trust (Oelschlager, 2024). Thus, detecting and explaining hallucinations is critical to LLM reliability.

In this paper, we propose a neurosymbolic framework, KEA (Kernel-Enriched AI) Explain, that not only detects semantic hallucinations in LLM-generated texts under both open-domain (general knowledge) and closed-domain (constrained context) conditions but also provides explanations of why the statements are deemed hallucinatory. By integrating symbolic AI methods with state-of-the-art neural techniques, our approach combines the strengths of both paradigms to address key limitations in existing methods. Specifically, we use graph kernels (Vishwanathan et al., 2010) to compare the structural similarity between knowledge graphs (KG), while semantic word embeddings (Almeida and Xexéo, 2019) are employed to cluster semantically similar labels and select relevant triples. For the open-domain problem, these techniques are combined with structured knowledge from Wikidata (Vrandečić and Krötzsch, 2014), a comprehensive and widely-used knowledge base, to form a robust detection system. We extract entities and relations from the LLM output and query these against the knowledge base, systematically identifying and explaining hallucinations based on discrepancies in structural and semantic similarity. For the closed-domain problem, we construct a KG based on both the LLM output and the provided context in place of the Wikidata knowledge. There is evidence that people prefer explanations that are

contrastive; that is, they explain why one outcome occurred rather than another plausible alternative (Miller, 2019). By providing *why* a given output was classified as a hallucination while also providing what fact would make it *not* a hallucination, KEA Explain offers a more explainable and intuitive solution.

The key contributions of this paper are as follows:

- Introduces the use of graph kernels over symbolic knowledge graphs to detect and explain hallucinations in LLM outputs.
- Presents a novel neurosymbolic framework to enable comparison of semanticity between entity and relation labels of a knowledge graph pair.
- Improves upon existing methods by introducing explainable classifications, which are driven by the symbolic structure of the graph-based representation.

## 2. Related Work

Hallucinations in LLMs stem from biased training data, lack of in-built logic, vague prompts, and insufficient data or overfitting (Perković et al., 2024). Common techniques to mitigate the production of hallucinations include fine-tuning and Retrieval-Augmented Generation (RAG) (Gao et al., 2023), which supplements LLM outputs with external data. However, RAG depends on prompt quality, potentially retrieving relevant but insufficient context (Guan et al., 2024). Given these limitations, hallucination detection is pertinent. Here, we review recent advances in hallucination detection.

### 2.1. LLM-based and Probabilistic Techniques

Many recently developed methods use LLMs or probabilistic measures to detect hallucinations. ChainPoll (Friel and Sanyal, 2023) aggregates votes from multiple LLM instances, but it is computationally intensive and lacks ground truth validation, so biases in the training data of the LLMs being used could propagate through. SelfCheckGPT (Manakul et al., 2023) estimates hallucination probability based on response entropy, assuming that domain knowledge leads to consistent outputs. While effective, both methods lack explainability and require multiple LLM queries, increasing computational cost. Belief Tree Propagation (BTPROP) (Hou et al., 2025) enhances detection by constructing a tree of logically related statements, analyzed via a hidden Markov model. This improves accuracy but is also resource-intensive, as tree expansion incurs exponential time complexity.

### 2.2. Knowledge Graphs for Hallucination Detection and Prevention

**Open Domain.** AlignScore (Zha et al., 2023) uses a unified information alignment function, a model designed to assess the alignment between two pieces of text, and is trained on 4.7M examples spanning seven language tasks, including paraphrasing, fact verification, and summarization. However, its reliance on synthetic training data introduces questions about real-world generalizability. Knowledge Graph-based Retrofitting (KGR) (Guan et al., 2024) reduces factual hallucinations in LLMs by refining outputs using KGs. Unlike methods that only retrieve facts based on user queries, KGR is able to extract, validate, and correct facts within the LLM's reasoning process, enabling it to catch inaccuracies generated during intermediate reasoning, however, entity detection noise can prevent accurate claim validation.

**Closed Domain.** FactAlign (Rashad et al., 2024) constructs a KG from a given source and generated text, and uses word embeddings to align individual claim triples with source triples to identify factual misalignments, categorizing them as hallucinations if the similarity is low. FactAlign differentiates between intrinsic hallucinations (distortions of source information) and extrinsic hallucinations (unsupported additions) by employing a contradiction score from a natural language inference (NLI) model. The approach achieves high accuracy scores without requiring training or fine-tuning. However, it excludes non-named entities in the KG, which reduces coverage, and encounters scalability challenges due to the computational overhead of computing pairwise similarities between each generated triplet and all source triplets. Additionally, the triple-level comparison methodology in Rashad et al.'s approach prevents wider context from the knowledge graph from contributing to similarity calculations, a limitation that our graph kernel-based approach overcomes. GraphEval (Sansford et al., 2024) decomposes LLM outputs into entities and relationships, validating them against the grounding context provided using NLI. On established benchmarks, it performs well on long outputs, but is limited to closed-domain settings, where the method is provided with the output alongside both the original prompt and the contextual information that was used to arrive at that output. This limits GraphEval's real-world applicability.

### 2.3. Summary
Across these studies, three common limitations emerge:

1. **Generalizability**: The majority of methods are designed to rely on specific open- or closed-domain conditions, but not both. In addition, many make use of synthetic training data, limiting applicability beyond their training domain.

2. **Lack of ground-truth**: Many of these methods use proxies for ground truth, such as measuring the entropy of LLM responses, or neural NLI models trained on synthetic data, introducing biases.

3. **Lack of explainability**: Neural methods alone are not transparent, making it difficult to understand why certain outputs are classified as hallucinations, degrading user trust.

### 3. Method
Our proposed method systematically detects and explains hallucinations in LLM outputs through comparison of KG representations. Initially, an LLM is employed to perform KG construction using the given text to obtain a KG representing its claims. For the open-domain problem, this claim KG is then paired with a ground truth KG derived from relevant triples within Wikidata, with both graphs capturing the relationships and attributes of the identified entities. For the closed-domain problem, the claim KG is compared with a KG constructed via the same LLM-based method applied to the provided context.

The KGs are then compared by using a graph kernel, which provides a numerical measure of the structural similarity between the KG pair (Shervashidze et al., 2011). In cases where the similarity score falls below a predetermined threshold, indicating a potential hallucination, an explanation is generated via a two-step analysis process. First, contradictory relations between the two KGs are identified based on embeddings of the triple entities. A modified graph edit distance algorithm is then used to identify the specific structural differences between the claim and ground-truth KGs. These differences are provided to an

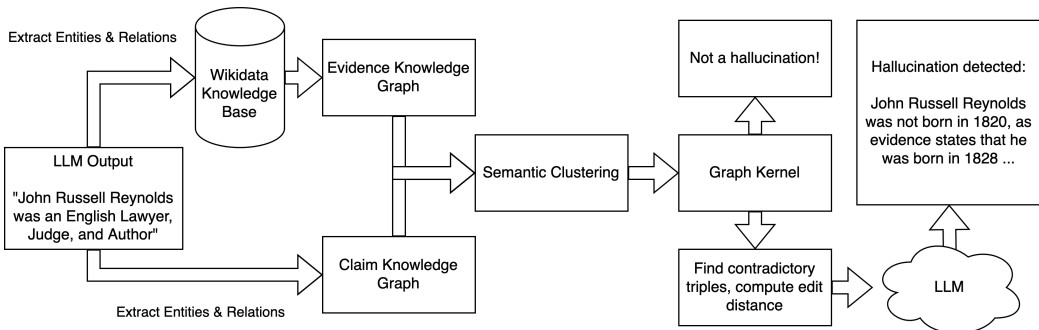

Figure 1: Visual depiction of the open-domain proposed method.

LLM to generate a contrastive explanation that explains why the original output qualifies as a hallucination, highlighting specific discrepancies between the model's claims and the established facts (Miller, 2019).

### 3.1. Knowledge Graph Construction

The process for converting unstructured textual data into a KG is split into three key stages (Sansford et al., 2024). 1) **Named Entity Recognition (NER)**: Identify atomic entities from the text, such as people, organisations, locations, dates, etc.; 2) **Coreference resolution**: Find all mentions in the text that refer to the same entity and relabel the detected entities accordingly; and 3) **Relation extraction**: Identify relations between the detected entities.

Large language models (LLMs) have become the standard tool for constructing knowledge graphs (KGs) in hallucination detection due to their ability to extract contextual features from text. AutoKG (Chen and Bertozzi, 2023) demonstrates their potential for automated KG generation. Methods like GraphEval (Sansford et al., 2024) enhance performance via in-context learning and chain-of-thought (CoT) prompting to guide entity and relation extraction. Building on this, our approach applies similar prompting strategies to constrain the LLM to task-specific knowledge and reduce hallucinated triples. We set temperature to 0 for deterministic responses and support modular upgrades to newer LLMs without modifying the core methodology. The full prompt is provided in Appendix D.

To evaluate LLM-generated KGs in open-domain tasks, we first retrieve ground-truth data from Wikidata. Using the Spacy Entity Linker (Gerber, 2024), detected entities are aligned with corresponding Wikidata entries, enabling SPARQL queries to extract relational triples between linked entities. These triples form the ground-truth KG for assessing the completeness and correctness of the LLM-generated KG. To enrich semantic depth, we also retrieve entity descriptions from Wikidata and Wikipedia, which are parsed using the same LLM-based method used to construct the claim KG. This added context improves coverage and strengthens comparison robustness.

### 3.2. Knowledge Graph Relation Selection

Because the context or ground-truth KG often contains a significant amount of information which is irrelevant to the claim KG, relation selection must be done to refine it so that the two KGs can be compared using the graph kernel. For this task, our method uses embeddings obtained from Sentence-BERT (SBERT) (Koroteev, 2021). SBERT provides

sentence embeddings that can be compared using cosine similarity. To obtain embeddings for KG triples, we concatenate the components into a string, e.g., the triple ("Albert Einstein", "was born in", "Ulm") is transformed into "Albert Einstein was born in Ulm". This concatenated string is then passed through SBERT to generate an embedding for the triple.

To select the most relevant triples from the context/ground-truth KG, we match each triple in the claim KG with the most semantically similar triple from the context/ground-truth KG. Relevance is determined by maximizing the cosine similarity between the embeddings of the current claim KG triple and a triple from the context/ground-truth KG. Specifically, the relevant triple $T_{context}$ from the context KG is identified as the one that maximizes the cosine similarity between its embedding $E_{context}$ and the embedding of the current claim KG triple $T_{claim}$:

$$T_{context}(T_{claim}) = \arg\max\left(\frac{E_{context} \cdot E_{claim}}{|E_{context}| \cdot |E_{claim}|}\right) \tag{1}$$

This process results in a refined ground-truth KG, of size at most equal to the size of the claim KG. The ground-truth KG now only contains the most relevant triples to the claim.

### 3.3. Semantic Clustering of Knowledge Graph Labels

Since graph kernels have no knowledge of the semantic meaning of graph labels, a mechanism for dealing with semantically similar but syntactically differing labels between graphs must be introduced. To achieve semantic comparison of labels, SBERT is used again to produce word embeddings for each node and edge label across both KGs. These embeddings are then clustered using an agglomerative hierarchical clustering algorithm. Cosine similarity is used as the cluster metric, with average linkage as the criterion (Ackermann et al., 2014). A distance threshold of 0.35 was chosen empirically, based on maximization of performance on hand-created tests as well as benchmarks covered in the Experiments section. The result allows for semantically similar labels to be clustered together under the same label, e.g., 'capital of France' and 'Paris' would both be grouped into the same cluster, and represented under the same label.

### 3.4. Graph Kernel Comparison of Knowledge Graphs

The next step is to compute a comparison score between the KG pair using the Weisfeiler-Lehman (WL) graph kernel (Shervashidze et al., 2011). The WL kernel is based on iteratively refining the node labels of each KG through neighborhood aggregation. Initially, each node in the graph is labeled with its own numerical identity or a basic feature. Then, for each iteration, the label of each node is updated by combining the labels of its neighbors. This process is repeated for a fixed number of iterations, and the inner product of the final high-dimensional feature representation vectors of each KG is used to compute a kernel score that reflects the structural similarity between the graphs. See Appendix A for more details.

The WL kernel is effective for comparing KGs because it can capture graph isomorphisms and structural patterns at different levels of abstractions (such as at the entity level, relation level, and subtree level), while taking into account the presence of varying node labels or relational differences. Thus, we can compare the KGs in a manner that accounts for both the topology of the graphs and the semantic relationships (labels) encoded within

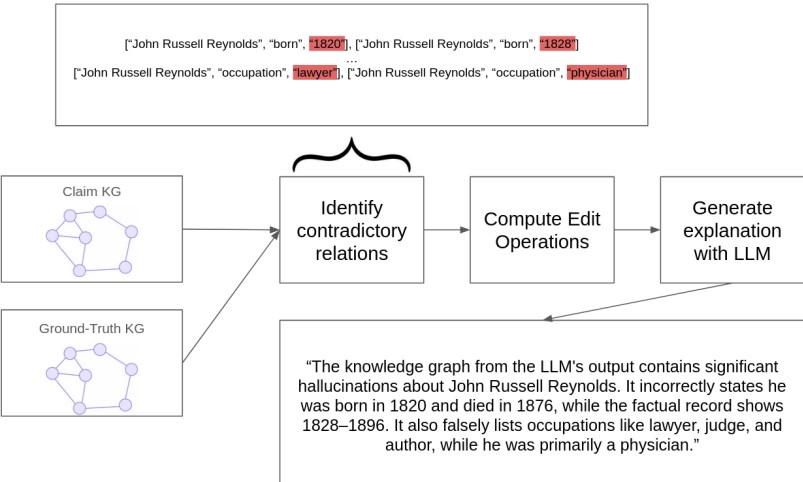

Figure 2: Explanation generation from the claim and ground-truth knowledge graphs.

them. This method is particularly effective in identifying subtle differences between KGs, which is important in the context of detecting hallucinations or inconsistencies in KG-based representations of text. Our implementation used the open-source graph kernel library, GraKeL (Siglidis et al., 2020). For our method, we set the number of iterations to five and normalize the output of the kernel to return a similarity score between zero and one.

### 3.5. Generation of Explanations

To generate contrastive explanations, we first identify pairs of contradictory relations across the two original KGs (prior to relation selection). These are pairs of triples where two of the three elements are semantically similar, but the third differs significantly. We determine similarity using word embeddings and thresholding on cosine similarity. For instance, ("France", "capital city", "Paris") and ("France", "capital", "Rome") form a contradictory pair, as the first two elements align semantically, while the third does not. This approach identifies key discrepancies between the graphs, rather than simply unrelated triples. Next, a simplified graph edit distance algorithm (Gao et al., 2010) determines the sequence of edge operations required to transform the claim KG's contradictory relations into those of the ground-truth KG. In the above example, the algorithm identifies the need to add "Paris" as France's capital and remove "Rome".

Finally, an LLM generates a natural language explanation of the detected hallucination based on these contradictory relations and edit operations. Unlike template-based methods, the LLM provides more flexible and detailed explanations. Rather than stating "Edge X is missing", it might generate: "The claim states 'X' between 'A' and 'B', but the ground-truth KG asserts 'A' and 'B' are related through 'Y'. This is incorrect." This enhances interpretability and helps pinpoint the hallucinations. An example of this process is shown in Figure 2.

### 4. Experiments

For evaluation of our method, we carried out three separate experiments. The first of which focuses on the closed-domain hallucination detection problem, where the detection method

| Method | SummEval | QAGS-C | Average Score |
|---|---|---|---|
| HHEM He et al. (2021) | 0.660 | 0.635 | 0.648 |
| GraphEval (HHEM) | 0.715 | 0.722 | 0.714 |
| TRUE Honovich et al. (2022) | 0.613 | 0.618 | 0.616 |
| GraphEval (TRUE) | 0.724 | 0.717 | 0.721 |
| TrueTeacher Gekhman et al. (2023) | 0.749 | 0.756 | 0.753 |
| GraphEval (TrueTeacher) | **0.792** | **0.781** | **0.787** |
| KEA Explain (Ours) | 0.761 | 0.711 | 0.736 |

Table 1: Comparison of balanced accuracy results for the SummEval and QAGS-C experiments. Bold indicates top-ranked, while underlined indicates second-ranked

has access to a specific, constrained context which is supplied alongside the LLM's generated content. The second experiment focuses on detection of open-domain hallucinations, where the detection method has access to external sources, such as a knowledge base as used in our method. Thirdly, we run an experiment to evaluate the efficacy of our method's generation of explanations of detected hallucinations.

In reporting the results of these experiments, we align with benchmark methods by using the same metrics for comparison. A comprehensive report of the results for all metrics (accuracy, balanced accuracy, precision, recall, and F1) is provided in Appendix C. Source code for this project can be viewed at https://github.com/Reih02/hallucination_explanation_graph_kernel_analysis.

### 4.1. Closed-Domain Hallucination Detection

Our evaluation employs two widely-used benchmarks for assessing closed-domain hallucination detection in the current literature. We compare our results to GraphEval (Sansford et al., 2024), a similar KG-based approach that enhances Natural Language Inference (NLI) models. GraphEval uses two benchmarks: SummEval (Fabbri et al., 2021), which evaluates 1600 summaries of 100 CNN/DailyMail articles with human ratings, and QAGS-C (Wang et al., 2020), derived from 235 CNN/DailyMail articles. A sentence is labeled hallucinatory if its average consistency score is below 0.6.

We empirically set graph kernel similarity thresholds at 0.15 (SummEval) and 0.5 (QAGS-C) for optimal balanced accuracy. Results (Table 1) show that our method outperforms four of six GraphEval methods on average, ranking just behind TrueTeacher, which benefits from synthetic data in fine-tuning, a bias risk we avoid due to the criticality of using a ground-truth source in hallucination detection. KEA Explain performs best on the SummEval task, with a balanced accuracy of 0.761 (second out of all methods). The performance in the QAGS-C task is slightly worse, coming in fourth overall, behind the three knowledge-graph enhanced methods. Overall, this shows that the performance of our method is comparable to existing closed-domain knowledge-graph based hallucination detection methods.

The ROC curves (Figure 3) visualize the performance across different graph kernel thresholds, with AUC values: 0.79 (SummEval) and 0.70 (QAGS-C). Both curves bend toward the top-left, indicating high True Positive Rates (TPR) with low False Positive Rates (FPR). In SummEval's ROC curve, a linear segment at higher thresholds suggests the classifier approaches random guessing beyond a certain threshold.

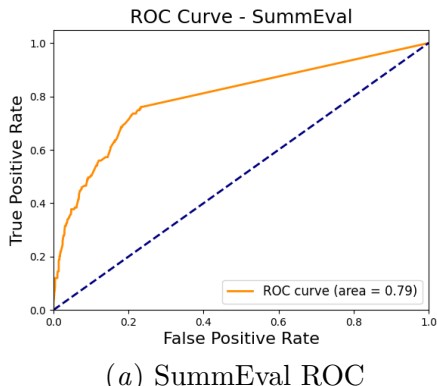 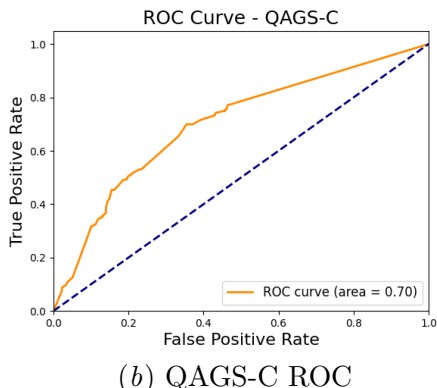

$(a)$ SummEval ROC $\qquad$ $(b)$ QAGS-C ROC

Figure 3: ROC Curves for our classifier on different benchmarks.

There is also a notable performance difference of KEA Explain across the two benchmarks. We hypothesize that this performance difference is due to QAGS-C's shorter sentences, which result in less informative KGs, making the graph kernel more sensitive to minor discrepancies between summaries and context.

### 4.2. Open-Domain Hallucination Detection

To assess our method's effectiveness in addressing open-domain hallucination detection, we utilized the WikiBio GPT-3 hallucination dataset. This dataset contains 238 GPT-3 (`text-davinci-003`)-generated passages that resemble Wikipedia-style content. Each passage is segmented into sentences and annotated as accurate, containing minor inaccuracies, or containing major inaccuracies. Sentences labeled with minor or major inaccuracies are considered hallucinatory, while accurate sentences are classified as consistent.

We report comparisons with SelfCheckGPT (Manakul et al., 2023) and AlignScore (Zha et al., 2023), two open-domain approaches, using the results reported by the authors of FactAlign (Rashad et al., 2024), a closed-domain method. We follow these papers by reporting Precision, Recall, and F1 scores, optimizing the graph kernel similarity threshold to 0.3 in order to maximize the F1 score. Our method achieved a similar F1 score to the other methods but had lower precision, indicating a tendency to classify non-hallucinated responses as hallucinations (Table 2). Despite this, our method outperformed the other benchmarks in recall, suggesting it is particularly effective at detecting true positives (hallucinations) in open-domain settings.

We hypothesize that the lower precision stems from limitations in the knowledge base querying process, particularly the inability to retrieve niche or highly-specific entities from Wikidata, increasing the likelihood of false positives. While this reliance on Wikidata leads to more false positives, it also provides a ground-truth reference for factual accuracy, which other methods lack (Manakul et al., 2023; Zha et al., 2023). Both SelfCheckGPT and AlignScore also showed lower precision compared to recall, indicating that handling of false positive predictions is a common challenge in open-domain hallucination detection.

Due to the class imbalance in the WikiBio dataset, we report a Precision-Recall curve (Figure 4). We observe an AUC of 0.77 for this PR curve. The curve shows an initial sharp

| Method | Precision | Recall | F1 |
|---|---|---|---|
| SelfCheckGPT | **0.843** | 0.917 | 0.879 |
| AlignScore | 0.809 | 0.981 | **0.886** |
| KEA Explain (Ours) | 0.734 | **0.984** | 0.841 |

Table 2: Comparison of Precision, Recall, and F1 results for the WikiBio dataset.

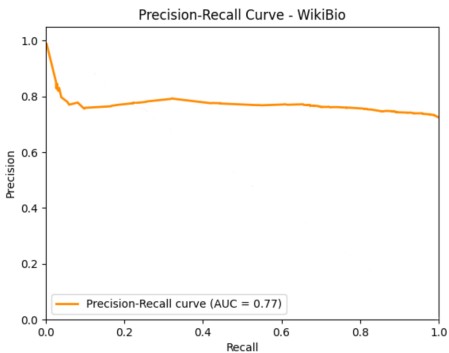

Figure 4: Precision-Recall Curve on the WikiBio benchmark.

precision drop as recall increases from 0 to 0.1 when the graph kernel threshold decreases. Precision then stabilizes around 0.75-0.8 across most recall values (0.1 to 0.9). With 73% of examples being hallucinatory (our positive class), this majority helps maintain relatively high precision throughout the curve. As thresholds decrease further, the model captures more true hallucinations while precision remains steady because correctly identifying the abundant positive class offsets the increasing false positives. The AUC of 0.77 reflects this balanced performance.

### 4.3. Hallucination Explanations

To evaluate the effectiveness of our method for generating explanations of detected hallucinations, we introduce a set of criteria for evaluating the "goodness" of an explanation inspired by the 'Explanation Goodness Checklist' provided by Johs (2024) (see Table 3). To conduct our evaluation, we randomly selected 20 entries (see Appendix B.2 for examples of generated explanations) with a consistency label $\leq 3$ from the SummEval dataset, in order to evaluate the "goodness" of our explanation generation system according to our defined evaluation criteria. The explanation is compared with the original article and the generated summary, in order to get a clear picture of how well the explanation covers the hallucination(s) present in the example. We focused on the closed-domain for the explanation evaluation due to ease of manual evaluation.

We defined three groups based on the consistency rating label of the LLM output for which the explanation was generated: Group 1 contains consistency ratings between 0 and 1; Group 2 ratings between 1 and 2; and Group 3 ratings between 2 and 3. In Group 1, the average rating was 4.85, while Group 2 received 4.15, and Group 3 scored 3.15. These results reveal a clear trend: as the consistency of the output increased (indicating a less-significant hallucination), explanation quality declined. This pattern held across all four evaluation criteria—accuracy and reliability, trustworthiness, detail, and completeness (Figure 5). Notably, while the accuracy and reliability of the explanations remained relatively close across the groups, the most significant decline in ratings was observed in the "detail" criterion. This suggests that, as the significance of the hallucination in the LLM's output decreased, more details were omitted from the explanation that could have enhanced its quality. We hypothesize that this effect is due to the decreased likelihood of finding relevant conflicting triples to guide the explanation generation process when the hallucinations are

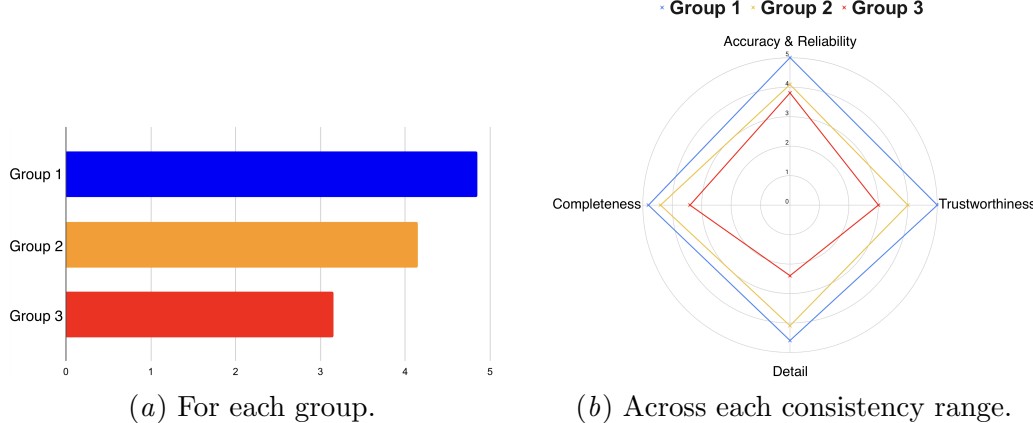

($a$) For each group.  ($b$) Across each consistency range.

Figure 5: Average ratings across all four features.

more sparse and nuanced. In this case, the explanation-generation process has less guidance as to exactly where the hallucination occurred, and crucial details could get left out of the generated explanation.

## 5. Discussion and Conclusion

In this paper, we presented a novel framework for detecting and explaining hallucinations in LLMs. By leveraging graph kernels and semantic clustering techniques, our approach identifies inconsistencies between LLM outputs and external knowledge bases or provided context. The introduction of explainable graph-based mechanisms addresses critical limitations in existing hallucination detection methods, offering both interpretability and robustness across open-domain and closed-domain tasks.

KEA Explain performs competitively with state-of-the-art approaches, achieving strong benchmark scores across both open- and closed-domain tasks. By utilizing a KG, it is able to store captured knowledge from the LLM's output and the ground-truth source in a relational format. This enables it to pinpoint discrepancies between KGs allowing it to supply the user with a contrastive explanation as to why the output was detected to be a hallucination. These explanations help users understand the discrepancies between generated outputs and factual knowledge.

The use of a graph kernel for KG comparison has further benefits over other hallucination detection methods. Rather than matching individual triples between two KGs such as is seen in FactAlign (Rashad et al., 2024) and GraphEval (Sansford et al., 2024), graph kernels enable us to take into account the neighboring structure of a subgraph at varying levels of abstraction (such as entity comparison, relation comparison, and sub-graph comparison) in order to arrive at a detailed comparison. This facilitates the capture of more structural information between the graphs and fills in the gaps where certain data is missing.

Still, the method has limitations, including sensitivity to domain-specific graph kernel thresholds and challenges in handling highly specific entities in open-domain settings. Future research directions include refining entity-linking mechanisms to improve precision, enhancing explanation generation for nuanced hallucinations, and exploring multi-source knowledge base integration to further strengthen robustness.

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

## Appendix A. Definition of the Weisfeiler-Lehman Graph Kernel

A graph kernel is a function $k : \mathcal{G} \times \mathcal{G} \to \mathbb{R}$ that measures the similarity between two graphs, where $\mathcal{G}$ denotes the set of graphs. Graph kernels are a type of kernel function used in machine learning to enable the application of algorithms, such as Support Vector Machines (SVMs), to structured data like graphs. They compute the similarity between graphs by comparing their structural and attribute-based features Vishwanathan et al. (2010).

Mathematically, a graph kernel can be expressed as an inner product in a high-dimensional feature space:

$$k(G_1, G_2) = \langle \phi(G_1), \phi(G_2) \rangle \tag{2}$$

Where $G_1$ and $G_2$ are graphs, and $\phi : \mathcal{G} \to \mathbb{R}^d$ is a feature mapping that embeds graphs into a $d$-dimensional vector space. These kernels provide a way to quantify how 'similar' two graphs are by capturing structural patterns at different levels of abstraction, such as common subgraphs, paths, and tree structures present across the pair.

In this project, we utilize the Weisfeiler-Lehman Subtree Kernel Shervashidze et al. (2011), which leverages the Weisfeiler-Lehman graph isomorphism test to generate subtree patterns for comparison. To define the Weisfeiler-Lehman Subtree Kernel, let us assume the following:

- $G$, $G'$ are two graphs

- $\Sigma_i \subseteq \Sigma$ are the set of letters that occur as node labels at least once in $G$ or $G'$ at the end of the $i^{th}$ iteration of the Weisfeiler-Lehman algorithm

- $\Sigma_0$ is the set of original node labels of $G$ and $G'$

- We define a map, $c_i : \{G, G'\} \times \Sigma_i \to \mathbb{N}$ such that $c_i(G, \sigma_{ij})$ is the number of occurrences of the letter $\sigma_{ij}$ in the graph $G$.

The Weisfeiler-Lehman subtree kernel on two graphs $G$ and $G'$ with $h$ iterations is then defined as the following:

$$k(G, G') = \langle \phi(G), \phi(G') \rangle \tag{3}$$

where

$$\phi(G) = (c_0(G, \sigma_{01}), ..., c_o(G, \sigma_{0|\Sigma_0|}), ..., c_h(G, \sigma_{h1}), ..., c_h(G, \sigma_{h|\Sigma_h|})) \tag{4}$$

and

$$\phi(G') = (c_0(G', \sigma_{01}), ..., c_o(G', \sigma_{0|\Sigma_0|}), ..., c_h(G', \sigma_{h1}), ..., c_h(G', \sigma_{h|\Sigma_h|})) \tag{5}$$

It can be shown that the Weisfeiler-Lehman Subtree Kernel can be computed in $\mathcal{O}(hm)$ time, where $h$ is the number of iterations and $m$ is the total number of edges across all graphs.

## Appendix B. Explanations

### B.1. Explanation Criteria Ratings

Table 3 shows the qualitative rating descriptions for the four explanation criteria: level of detail, completeness of explanation, accuracy and reliability, and trustworthiness.

| Criteria | Rating descriptions |
|---|---|
| **Level of Detail** | 1: Too brief, lacks necessary detail. 
 2: Somewhat detailed 
 3: Adequate detail, could be deeper. 
 4: Well-detailed, covers most aspects. 
 5: Comprehensive, in-depth. |
| **Completeness of Explanation** | 1: Incomplete, misses major aspects. 
 2: Some key components missing. 
 3: Mostly complete, with minor gaps. 
 4: Almost complete. 
 5: Fully complete, covers all aspects. |
| **Accuracy and Reliability** | 1: Inaccurate, major errors. 
 2: Some inaccuracy or missing context. 
 3: Mostly accurate, small errors. 
 4: Very accurate, minor issues. 
 5: Fully accurate and reliable. |
| **Trustworthiness** | 1: No supporting evidence. 
 2: Lacking sufficient evidence. 
 3: May lack evidence. 
 4: Well-supported claims. 
 5: Trustworthy, solid evidence. |

Table 3: Criteria for rating generated explanations.

## B.2. Example Explanations

### B.2.1. SAMPLE 1

The knowledge graph generated from the LLM's output contains several key inaccuracies that highlight its hallucinatory nature. Notably, it incorrectly asserts a relationship between 'Space Invaders' and the year '1970', suggesting that the game was developed in that specific year. However, this claim is not supported by factual data, as the actual development period is more accurately described as occurring in the late 1970s. This discrepancy indicates a significant misunderstanding or misrepresentation of the timeline associated with the game's creation. Furthermore, the absence of the correct relationship linking 'Space Invaders' to the late 1970s in the LLM's output further emphasizes the inaccuracies present in the generated knowledge graph. Together, these false claims illustrate how the LLM's output diverges from established facts, leading to a misleading representation of the game's historical context.

### B.2.2. SAMPLE 2

The knowledge graph generated from the LLM's output contains several significant hallucinations that misrepresent the facts surrounding Ben Stokes and his experiences in cricket. Firstly, the LLM incorrectly states that Stokes "broke his neck" during the Ashes series, which is a serious misrepresentation; in reality, he broke his wrist after punching a locker, an incident that occurred the previous year. This error not only alters the nature of the injury but also misplaces the context of his struggles, as the original text emphasizes his need to manage his aggression rather than suggesting a severe injury like a broken neck.

Additionally, the LLM's output inaccurately implies that Stokes is currently at the Kensington Oval, when in fact, the context suggests he is back in the England team and preparing for a match in Barbados. The relationship between Kensington Oval and England is also misrepresented, as the original context indicates a more nuanced connection, specifically that the Oval is a venue where Stokes has faced challenges. These inaccuracies collectively distort the narrative of Stokes's character and his journey, leading to a misleading portrayal of his situation in the England cricket team.

## Appendix C. Full Hallucination Detection Experiment Results

| Benchmark | Accuracy | Balanced Accuracy | Precision | Recall | F1 |
|---|---|---|---|---|---|
| SummEval | 0.782 | **0.761** | 0.276 | 0.736 | 0.401 |
| QAGS-C | 0.738 | **0.711** | 0.492 | 0.656 | 0.562 |
| WikiBio | 0.730 | 0.523 | 0.734 | 0.984 | **0.841** |

Table 4: Full hallucination-detection experiment results of our method across each benchmark. Metrics are marked in bold where they were the main focus of the benchmark comparison, and hence were optimised for in the graph kernel threshold selection process.

## Appendix D. LLM Prompt
The following is the detailed prompt used for generating knowledge graphs from unstructured text. It was heavily inspired by Sansford et al. (2024):

```
messages=[
    {"role": "system", "content": "You are an expert at creating
        knowledge graphs based on text.\n"
     "You will receive two separate pieces of text, and you must
        perform the following steps on each piece of text:\n"
     "1. Entity detection: Select key and crucial entities from the
        text. Keep these entities short and concise and skip less
        important details of the text\n"
     "2. Coreference resolution: Across both texts, ensure that you
        use the same entity name for the same concept. For example,
        \"He\" may actually refer to the entity \"Peter\". Also
        apply this step between texts, so that the two knowledge
        graphs can be compared as easily as possible without
        confusion.\n"
     "3. Relation extraction: Identify semantic relationships
        between detected entities. These relationships should be
        encapsulated as a simple and concise relation such as \"
        began in\", or \"will simulcast\", for example.\n"
     "4. Knowledge Graph refinement: Once the two knowledge graphs
        have been created, try to ensure that similar triples
        between the two texts / knowledge graphs are represented the
         same way, to avoid confusion. For example, if two different
         entities refer to a similar event or concept, relabel them
        to be the same across the two knowledge graphs.\n\n"
     "Format your response as a JSON object that can be directly
        parsed without any edits to your response. This means that
```

```
        you are not allowed to include any text not part of the
        knowledge graphs.\n"
    "In the JSON object, one element should be the knowledge graph
        for the first text, and another element should be the
        knowledge graph for the second text.\n"
    "Each knowledge graph should be a list of triples, with each
        triple being a python list of the form [\"Peter\", \"height
        \", \"180cm\"].\n\n"
    "See below for some examples:\n\n"
    "EXAMPLE 1:\n"
    f"TEXT1: \n{sample_text1}\n\nTEXT2:\n{sample_text2}\n\n"
    "YOUR OUTPUT:\n"
    "{\n"
    "    \"knowledge_graph1\": [\n"
    "        [\"A&E Networks\", \"will simulcast in 2016\", \"Roots
        \"],\n"
    "        [\"Roots\", \"premiered in\", \"1977\"],\n"
    "        [\"Roots\", \"ran for\", \"four seasons\"],\n"
    "        [\"Roots\", \"instance of\", \"miniseries\"],\n"
    "        [\"Roots\", \"followed\", \"Kunta Kinte\"],\n"
    "        [\"Kunta Kinte\", \"was sold into\", \"slavery\"],\n"
    "        [\"Kunta Kinte\", \"was a\", \"free black man\"]\n"
    "    ],\n"
    "    \"knowledge_graph2\": [\n"
    "        [\"Roots\", \"one of the\", \"biggest TV events of all
        time\"],\n"
    "        [\"Roots\", \"had a staggering audience of\", \"over 100
         million viewers\"],\n"
    "        [\"Roots\", \"being\", \"reimagined for new audiences
        \"],\n"
    "        [\"Roots\", \"was about\", \"an African-American slave
        and his descendants\"],\n"
    "        [\"Roots\", \"premiered\", \"1977\"]\n"
    "    ]\n"
    "}\n\n"

    "EXAMPLE 2:\n"
    f"TEXT1: \n{sample_text3}\n\nTEXT2:\n{sample_text4}\n\n"
    "YOUR OUTPUT:\n"
    "{\n"
    "    \"knowledge_graph1\": [\n"
    "        [\"ISIS\", \"released\", \"more than 200 Yazidis\"],\n"
    "        [\"Yazidis\", \"are\", \"minority group\"],\n"
    "        [\"ISIS\", \"released\", \"children and elderly Yazidis
        \"],\n"
    "        [\"Peshmerga commander\", \"said\", \"freed Yazidis are
        released\"]\n"
    "    ],\n"
    "    \"knowledge_graph2\": [\n"
    "        [\"ISIS\", \"released\", \"more than 200 Yazidis\"],\n"
    "        [\"Yazidis\", \"are\", \"minority group\"],\n"
```

```
    "        [\"Yazidis\", \"killed and displaced by\", \"ISIS\"],\n"
    "        [\"ISIS\", \"released\", \"children and elderly Yazidis
    \"],\n"
    "        [\"Peshmerga commander\", \"said\", \"freed Yazidis are
    released\"]\n"
    "        [\"Peshmerga\", \"received\", \"freed Yazidis\"],\n"
    "        [\"Peshmerga\", \"sent freed Yazidis to\", \"Irbil\"],\n
    "
    "        [\"Arab tribal leaders\", \"helped coordinate\", \"
    release of Yazidis\"]\n"
    "  ]\n"
    "}"},
    {"role": "user", "content": f"TEXT1: \n{text1}\n\nTEXT2:\n{
    text2}"}
]
```

