# OpenReview forum: "KEA Explain: Explanations of Hallucinations using Graph Kernel Analysis"
_nesyconf.org/NeSy/2025/Conference — NeSy 2025 Poster_

### Official Review · Reviewer_vLbg · 2025-04-04
**Interesting methodology, lacking impact and comparative justification**

**Rating:** 4
**Confidence:** 3

**Review:**

The paper proposes a neurosymbolic framework, namely Kernel-Enriched AI (KEA), to detect hallucinations in LLM outputs in both open-domain and closed-domain scenarios. This is achieved by comparing the structural properties of knowledge graphs (KGs) constructed from LLM outputs with ground truth KGs extracted from reliable sources, leveraging the concept of graph kernels.

The proposed framework is also explainable, as it can provide justifications for why a sentence is classified as hallucinatory and identify inconsistencies between the ground truth and the generated text. More specifically:

1. An LLM is used to construct a KG from a generated output;

2. The same LLM constructs a KG from a source: either a provided context (in the closed-domain setting) or a relevant Wikidata source (in the open-domain setting);

3. The two KGs are compared using a graph kernel, yielding a numerical similarity score; if this score falls below a predefined threshold, the output is classified as hallucinatory;

4. Contradictions between the KGs are identified by computing embeddings for triple entities;

5. Specific structural differences are detected using a graph edit distance algorithm, which is then used to generate fine-grained explanations highlighting the differences between the "ground KG" and the "claim KG".

While the procedure offers an interesting perspective on improving the reliability of LLMs through KG support, the authors ultimately fail to present a convincing argument for why this approach should be preferred over other similar, explainable methods (e.g., FactAlign). Additionally, the related work section would benefit from a more in-depth discussion, especially regarding how KEA improves upon or differs from existing methods.

Strengths:

* The exploration of graph kernels as a tool for hallucination detection in LLMs is novel and promising;

* The procedure and underlying algorithms are clearly described;

* The paper is well-structured and easy to follow.

Weaknesses:

* The results are not particularly compelling, and the authors do not provide a strong rationale for their method beyond explainability;

* The evaluation of explainability is limited to 20 randomly selected samples, which does not sufficiently demonstrate the effectiveness of the explanation mechanism;

* The related work section lacks depth, particularly in its discussion of competing approaches like FactAlign, and does not adequately articulate the limitations KEA is intended to address.

Questions:

1. This approach—and similar methods relying on an intermediate representation step (Text → KG)—seems to suffer from an inherent limitation: KG construction itself could be flawed, producing a structured representation that does not faithfully reflect the original text. This undermines the reliability of the entire process. Did the authors consider this issue? Were any specific techniques employed to assess or constrain the fidelity of the generated KGs to the input text?

2. Why are the FactAlign results missing in table 2?


Comments:

In the related work section, it was not immediately clear how the cited methods suffer from generalizability issues (as claimed in Section 2.3), particularly in relation to training data or domain dependence. The authors should provide a more concrete justification for this point in the discussion.

**Anonymity:**

Remain anonymous

---

### Official Review · Reviewer_vqto · 2025-04-07

**Rating:** 8
**Confidence:** 4

**Review:**

The paper introduces "KEA (Kernel-Enriched AI) Explain," a novel neurosymbolic framework designed to detect and explain hallucinations in Large Language Model (LLM) outputs. The framework operates by constructing knowledge graphs (KGs) from LLM-generated text and comparing them with ground truth KGs derived from Wikidata (for open-domain tasks) or the provided context (for closed-domain tasks). The core of the approach lies in using graph kernels to measure the structural and semantic similarity between these KG pairs. When a low similarity score indicates a potential hallucination, "KEA Explain" generates contrastive explanations by identifying contradictory relations and structural differences, ultimately using an LLM to articulate these discrepancies in natural language.

The novel application of graph kernels and the integration of symbolic and neural techniques offer an advantage in terms of interpretability and robustness. The generation of contrastive explanations is a particularly valuable feature. The paper demonstrates competitive empirical results, addressing the identified weaknesses, such as the sensitivity to thresholds, limitations in open-domain entity linking, and the quality of explanations for nuanced hallucinations.
Improving the justification for certain empirical choices, providing a discussion on computational cost, detailing the mathematical aspects, assessing the robustness of the single components, and strengthening the evaluation of explanation quality are key areas for improvement. By addressing these points, the authors can further solidify the contribution of KEA Explain.

**Anonymity:**

Remain anonymous